# Representation Learning for Dynamic Functional Connectivities via Variational Dynamic Graph Latent Variable Models

**DOI:** 10.3390/e24020152

**Published:** 2022-01-19

**Authors:** Yicong Huang, Zhuliang Yu

**Affiliations:** College of Automation Science and Technology, South China University of Technology, Guangzhou 510641, China; 201920116424@mail.scut.edu.cn

**Keywords:** neural latent variable models, dynamic functional connectivities, variational information bottleneck, dynamic graphs

## Abstract

Latent variable models (LVMs) for neural population spikes have revealed informative low-dimensional dynamics about the neural data and have become powerful tools for analyzing and interpreting neural activity. However, these approaches are unable to determine the neurophysiological meaning of the inferred latent dynamics. On the other hand, emerging evidence suggests that dynamic functional connectivities (DFC) may be responsible for neural activity patterns underlying cognition or behavior. We are interested in studying how DFC are associated with the low-dimensional structure of neural activities. Most existing LVMs are based on a point process and fail to model evolving relationships. In this work, we introduce a dynamic graph as the latent variable and develop a Variational Dynamic Graph Latent Variable Model (VDGLVM), a representation learning model based on the variational information bottleneck framework. VDGLVM utilizes a graph generative model and a graph neural network to capture dynamic communication between nodes that one has no access to from the observed data. The proposed computational model provides guaranteed behavior-decoding performance and improves LVMs by associating the inferred latent dynamics with probable DFC.

## 1. Introduction

Recent progress in invasive recording technologies, such as high-density microelectrode arrays [1,2,3] and optical fibers [4], allows access to large-scale neural population activities with the precision of single neurons. While these data are usually high-dimensional [5,6], the literature has shown that dimensionality-reduction approaches and generative models can faithfully explain population spike activities [7] and infer single-trial neural firing rates [8] with stable low-dimensional latent dynamics. At present, with rapid developments in machine learning and deep learning, the community has proposed several latent variable models (LVMs) with better efficiency and performance to extract the low-dimensional structure [9,10,11]. They bring novel insights into neuroscience [12,13] and facilitate the development of brain–computer interfaces [14].

However, existing LVMs consider the latent dynamics as abstract trajectories in the vector space without neurophysiological meaning. We are interested in how the coordination of different brain functions produces such low-dimensional trajectories. In other words, can we specify the exact meaning for each dimension of the trajectories based on neurophysiology? This question may help with understanding how cognitive or behavioral processes emerge. Recently, advances in network neuroscience [15,16] suggest that dynamic functional connectivities (DFC) may be a more accurate representation of functional brain networks [17,18,19,20]. This refers to dynamic relationships between distributed signals. DFC may be the mechanism for the coordination of activity between different neural networks to accomplish a complex task. Hence, it may also be a proper explanation for the latent dynamics discovered by LVMs. Nevertheless, while LVMs assume that latent dynamics follow a point process, DFC involve modeling dynamic relationships between variables. The community needs a more general framework to associate LVMs with DFC.

In this work, we improve LVMs and bridge LVMs and DFC by drawing on recent progress in graph representation learning and graph generative models [21], proposing a Variational Dynamic Graph Latent Variable Model (VDGLVM). We first generalize LVMs with the variational information bottleneck [22] framework to enable the design of non-Euclidean latent variables. We then leverage a dynamic graph as the latent variable to model population spiking and behavior simultaneously. The inferred evolution of relationships between nodes in the inferred dynamic graph can be associated with DFC implicitly. Thus, our model improves LVMs via specifying a probable explanation for the latent dynamics based on DFC. We evaluate our framework with real-world neural data. The results show that our method can not only decode behaviors with high performance but also identify simple yet interpretable representations that are informative of the task.

## 2. Background and Related Work

In this section, we describe the notations used in the present work, and we provide some background and related work on latent variable models, dynamic functional connectivity, and graph neural networks.

### 2.1. Latent Variable Models (LVMs)

We consider recorded spike count data from *N* neurons at *T* time steps, which results in an observation matrix $X\in N^{N\times T}$, with elements $x_{i,t}$ denoting the spike count of neuron i∈{1,…,N} at time t∈{1,…,T}, and the corresponding behavior $Y\in R^{B\times T}$, with *B* as the dimension of the behavior. The literature assumes that the spikes are generated from an inhomogeneous Poisson process based on an underlying non-negative value firing rate. Most LVMs try to find a lower-dimensional trajectory $Z\in R^{L\times T}$ with L⩽N, which explains the observed neural data. In this paper, we focus on dynamical models for behavior decoding. Usually, this is accomplished within the auto-encoder framework. Different choices of the encoder and the decoder may result in different LVMs, including the Gaussian Process (GP) [23,24,25,26,27], linear dynamical system (LDS) [28,29], multi-layer perceptrons (MLP) [30,31], recurrent neural networks (RNN) [8,26], etc. Moreover, since most LVMs are generative models, some approaches extract complex latent structures by carefully designing constraints on the latent space. For example, pi-VAE explicitly uses brain states as labels and models them with the latent variable in a supervised manner [31] and Swap-VAE leverages self-supervised learning techniques to decompose the latent space [30].

In this work, we consider the dynamic communications of nodes as the underlying dynamics. While existing LVMs only use a point process without exact neurophysiological meanings, our model introduces DFC to explain the latent dynamics. A slight yet significant difference between the proposed model and LVMs mentioned above is that we train our model to decode behavior in a supervised manner. This fits more closely with the information-based framework described below.

### 2.2. Dynamic Functional Connectivities (DFC)

Recently, the focus of the systems neuroscience community tends to shift from the analysis of the static correlation between signals to the study of dynamic relationships between different brain functions evolving with time [32]. The communications are often referred to as dynamic functional connectivities (DFC). To capture undirectional statistical dependencies between different neural signals, a common approach is investigating the correlations based on sliding windows [33,34,35,36]. To model directional dependencies, the most frequently used method is to estimate Granger causality [37,38,39]. Since DFC can be conveniently described using a dynamic graph, some works on optimizing the dynamic graph structure based on graph measures or designed constraints have been developed [40,41,42]. These approaches have been applied on modeling DFC based on functional magnetic resonance imaging (fMRI) [36], electroencephalograms [43,44], and neurons [42].

For these methods of studying DFC, the nodes are directly defined in the data, and the challenge accurately estimates the correlations or edges based on the data. However, to associate LVMs with DFC, the nodes and node information for determining the edges are not available. To bridge this gap and reveal a probable neurophysiological meaning to improve LVMs, the present work extends LVMs and explores a computational model to infer a dynamic graph that can be associated with DFC.

### 2.3. Graph Neural Networks (GNNs)

A graph G=(V,E) is defined by its node set $V=\{v_{1},v_{2},\ldots ,v_{\left|V\right|}\}$ and edge set E⊆V×V. The numbers of nodes and edges are denoted by |V| and |E|, respectively. The structure of the graph can be described by an adjacency matrix, $A\in {\left\{0,1\right\}}^{\left|V|\times |V\right|}$. We also have d-dimensional features for each node in the graph, denoted as $X_{g}\in R^{|V|\times d}$.

GNNs are representation learning neural networks for graph-structured data [21,45,46]. They have achieved remarkable success in different areas, such as molecular graph generation [47], brain connectome analysis [48], etc. GNNs leverage a message-passing [49] procedure to obtain a representation for each node $v_{i}$. It first aggregates features of its neighbor nodes $N_{i}$ to obtain a message $m_{i}$ for $v_{i}$, with a multi-set function AGGREGATE as:(1)$$m_{i}=AGGREGATE\left(\left\{h_{j},v_{j}\in N_{i}\right\}\right),$$
where $h_{j}$ is the intermediate representation of node $v_{j}$. Second, GNNs update the representation for each node $v_{i}$ with the message via a function COMBINE as:(2)$$h_{i}=COMBINE\left(h_{i},m_{i}\right).$$GNNs repeat these two steps to iteratively aggregate neighbor information to obtain better node representations. For graph-level representation learning, GNNs use an additional multi-set function READOUT on the node embeddings to have a graph representation vector $h_{g}$:(3)$$h_{g}=READOUT\left(h_{k},v_{k}\in V\right).$$

In this work, we define DFC as the latent variable instead of constructing edges on nodes predefined on the data. Thus, we leverage graph generative models to infer a dynamic graph as the latent variable. We also use GNNs to map the dynamic relationships between nodes and behavior.

## 3. Methodology

In this section, we introduce VDGLVM. We start by introducing the learning objectives of LVMs based on the variational information bottleneck framework to understand LVMs at a higher level. Within the framework, we build a generative model to utilize a dynamic graph as the latent variable. VDGLVM inherits the expressiveness of deep neural networks and attempts to interpret neural latent dynamics by LVMs based on DFC.

### 3.1. Variational Information Bottleneck (VIB)

The information-theoretical framework explains deep neural networks as an information bottleneck [50]. A network is optimized to find the representation mapping that maintains the maximum mutual information between the input and output [51]. The literature has shown that deep neural networks that constrain information from the input to an intermediate representation tend to be more robust [22].

For LVMs, we find the latent point process Z by maximizing the mutual information between the latent variable Z and the behavior variable Y, restricting the mutual information between the spikes variable X and the latent variable Z with a constant $I_{c}$, resulting in a constrained optimization problem:(4)$$\begin{array}{cc} \max & MI(Z;Y)\\ s.t. & MI\left(X;Z\right)⩽I_{c}, \end{array}$$
where the exact form of MI(Z;Y) is:(5)$$\begin{array}{cc} MI(Z;Y) & =E_{p(Z,Y)}\left[\log\frac{p(Z|Y)}{p(Z)}\right]\\ & =E_{p(Z,Y)}\left[\log\frac{p(Y|Z)}{p(Y)}\right]. \end{array}$$We can be further formulate the objective by introducing a Lagrange multiplier β>0 as:(6)min−MI(Z,Y)+βMI(X,Z).This suggests that we can tune β to find a suitable bottleneck for deep neural networks to restrict the mutual information between X and Z. However, the direct optimization of the above objective function is intractable since mutual information is notoriously difficult to compute when we only have access to samples but not the exact distributions.

VIB [22] solves this by maximizing a lower bound of MI(Z,Y) and minimizing an upper bound of MI(X,Z) simultaneously. The key is to use a variational distribution q(Y|Z) to approximate the intractable conditional distribution p(Y|Z). This approximation indicates that KL(p(Y|Z)||q(Y|Z))⩾0, where KL(·||·) is the Kullback–Leibler divergence. Therefore, we have:(7)$$\begin{array}{cc} MI(Z;Y) & =E_{p(Z,Y)}\left[\log\frac{p(Y|Z)}{p(Y)}\right]\\ \; & ⩾E_{p(Z,Y)}\left[\log\frac{q(Y|Z)}{p(Y)}\right]\\ \; & =E_{p(Z,Y)}\left[\log q(Y|Z)\right]+H\left(Y\right), \end{array}$$
where H(Y) is the differential entropy of the output variable Y. We neglect this term in optimization since it is irrelevant to the model once given the data. This bound is tight when q(Y|Z)=p(Y|Z). Based on the graphical model of the information bottleneck principle, we have a lower bound of MI(Z;Y) as:(8)$$\begin{array}{c} MI\left(Z;Y\right)⩾\int_{X}\int_{Y}\int_{Z}p\left(X,Y\right)p\left(Z|X\right)\log q(Y|Z)dXdYdZ. \end{array}$$Similarly, an upper bound of MI(X;Z) is:(9)$$\begin{array}{c} MI\left(X;Z\right)⩽E_{p(X)}\left[KL(p(Z|X)||q(Z))\right], \end{array}$$
where q(Z) is a variational approximation of the latent distribution p(Z). The bound is tight when q(Z)=p(Z). Using the following empirical data distribution,
(10)$$p\left(X,Y\right)=\frac{1}{N_{s}}\sum_{n=1}^{N_{s}}\delta_{X^{\left(i\right)}}\left(X^{\left(i\right)}\right)\delta_{Y^{\left(i\right)}}\left(Y^{\left(i\right)}\right),$$
where $N_{s}$ is the number of samples and δ(·) is the Dirac delta function, we have the objective for LVMs based on VIB when the latent variable is a point process:(11)$$\begin{array}{cc} \min\frac{1}{N_{s}}\sum_{n=1}^{N_{s}} & \left[-\int_{Z}p\left(Z|X^{\left(i\right)}\right)\log q(Y^{\left(i\right)}|Z)dZ+\beta KL(p\left(Z|X^{\left(i\right)}\right)||q\left(Z\right))\right], \end{array}$$
where p(Z|X) is the encoder and q(Y|Z) is the decoder. The first term is the likelihood of output Y, given latent variable Z, inferred from input X. In this work, Y is the behaviors of the subject, and thus, we formulate this term based on a Gaussian distribution. The second term is regarded as a regularization term for the latent variable Z, given the prior distribution q(Z).

### 3.2. Dynamic Graphs as Dynamic Functional Connectivities

Most LVMs study neural activities in a particular region based on the behavior, while the behavior may be completed as a complex cognitive process involving communications between different brain regions. However, for LVMs, nodes and edges for defining a graph are not available on the data, and thus, one needs to infer them, given the data. This can be formulated as inferring the dynamic relationships between variables that are not defined in the observed data. Meanwhile, graphs are suitable to model dependencies. Recent progress on graph-based deep learning significantly promotes graph modeling both in performance and efficiency. We then wonder whether a graph is feasible as the latent variable of LVMs, such that it corresponds to DFC and provides a probable explanation for the latent dynamics based on graph generative models.

To simplify, we assume that the dynamic relationships faithfully capture the evolving nature of the observations. The present work aims to capture pair-wise dependencies between variables and emphasize the dynamic relationships in generating neural activities and behaviors. Thus, we do not consider node dynamics, but assume that the dynamic relationships already summarize all necessary information about the data. The nodes only serve as labels for the variables we are interested in.

Let $G_{t}=\left(V,E_{t}\right)$ be a dynamic graph, described by its dynamic adjacency matrix $A_{t}$ and static node features $X_{g}$, and let $G=\{G_{1},\ldots ,G_{T}\}$ be a graph process. Our objective is to replace the point process Z with a graph process G within the VIB framework. However, it is intractable to directly solve the problem due to the discrete nature of graphs. We start by assuming that the graph is a Gilbert random graph [52], such that every possible edge of a graph is conditionally independent of each other. Let $e_{ij}∼Bernoulli(\theta_{ij})$ be the binary variable indicating whether the edge (i,j) exists, where $\theta_{ij}$ is the probability that edge (i,j) occurs in a graph *G*. Based on the theory of Gilbert random graphs, the distribution of random graph variable *G* can be factorized as:(12)$$p\left(G\right)=\prod_{(i,j)\in E}p\left(e_{ij}\right).$$

We relax the edge weights from binary variables to continuous variables in the range (0,1), such that we can optimize the objective based on gradient descent efficiently. Specifically, we give a weight ${\hat{e}}_{ij}$ to every edge (i,j) to reformulate *G* as a smoothed-graph variable $\hat{G}$. The edge weights can be summarized in a modified adjacency matrix $\hat{A}\in {\left[0,1\right]}^{n\times n}$. Therefore, with a smoothed-graph process $\hat{G}=\left\{{\hat{G}}_{1},\ldots ,{\hat{G}}_{T}\right\}$, the objective function becomes:(13)$$\begin{array}{cc} \min_{}\frac{1}{N_{s}}\sum_{n=1}^{N_{s}} & \left[-\int_{\hat{G}}p\left(\hat{G}|X^{\left(i\right)}\right)\log q(Y^{\left(i\right)}|\hat{G})d\hat{G}+\beta KL(p\left(\hat{G}|X^{\left(i\right)}\right)||q\left(\hat{G}\right))\right], \end{array}$$
where $p(\hat{G}|X^{\left(i\right)})$ is a sequential graph generative model serving as the encoder, and $q(Y^{\left(i\right)}|\hat{G})$ is a sequential graph neural network serving as the decoder.

### 3.3. VDGLVM

In the VIB framework, we propose the use of dynamic graphs as the latent variable to study the relationship dynamics between static variables with VDGLVM. The architecture of VDGLVM is shown in Figure 1. The network is comprised of 4 layers. The encoder $p(\hat{G}|X^{\left(i\right)})$ includes the encoding block, extracting features from neural population spikes, and the graph generative model, generating graphs. The decoder $q(Y^{\left(i\right)}|\hat{G})$ includes the graph neural network, which performs graph-level representation learning, and the output layer, which maps graph embeddings to output behavior.

#### 3.3.1. Encoder

We instantiate $p(\hat{G}|X^{\left(i\right)})$ based on dynamical models. To capture nonlinear, non-Markovian, long short-term time-dependent dynamics, we utilize the RNN to extract features. At each time step *t*, RNN reads the spikes observation $x_{t}$ and updates its hidden state $h_{t}\in R^{H}$ by:(14)$$h_{t+1}=RNN_{\theta }\left(x_{t},h_{t}\right),$$
where $RNN_{\theta }$ is a deterministic non-linear transition function parameterized by θ. In practice, the vanilla RNN may suffer from gradient issues; it becomes hard to train and leads to performance degeneration. Thus, normally, $RNN_{\theta }$ is implemented via long short-term memory (LSTM) or a gated recurrent unit (GRU). It models the dynamic graph ${\hat{G}}_{t}$ by parameterizing a factorization of the joint sequence probability distribution as a product of conditional probabilities, such that:(15)$$\begin{array}{cc} p({\hat{G}}_{1},\ldots ,{\hat{G}}_{T}) & =\prod_{t=1}^{T}p\left({\hat{G}}_{t}|{\hat{G}}_{<t}\right)\\ p({\hat{G}}_{t}|{\hat{G}}_{<t}) & =p({\hat{G}}_{t}|h_{t}), \end{array}$$
where we instantiate $p_{\phi }\left({\hat{G}}_{t}|h_{t}\right)$ as a graph generative model parameterized by ϕ, which maps the RNN hidden state $h_{t}$ to a probability distribution of the graph variable ${\hat{G}}_{t}$, which can be factorized as:(16)$$p\left({\hat{G}}_{t}\right)=\prod_{i=1}^{n}\prod_{j=1}^{n}p\left({\hat{e}}_{ij_{t}}\right).$$To compute egde weights, we first use *n* MLPs to generate *n* Gaussian latent variables $v_{i_{t}}\in R^{d}$, summarized in $V_{t}\in R^{n\times d}$, from the RNN hidden state, as:(17)$$\begin{array}{cc} p\left(v_{i_{t}}|h_{t}\right) & =N\left(\mu_{v_{i_{t}}},\sigma_{v_{i_{t}}}^{2}\right)\\ \left[\mu_{v_{i_{t}}},\sigma_{v_{i_{t}}}^{2}\right] & =MLP_{i}\left(h_{t}\right), \end{array}$$
where, in practice, we share the parameters of $MLP_{i}$, except for the last layer. We apply the reparameterization trick to optimize the objective function with gradient-based methods. Specifically, we sample ϵ from a standard normal distribution, and then we generate ${v_{i}}_{t}$ via:(18)$$\begin{array}{cc} \epsilon & ∼N(0,I)\\ v_{i_{t}} & =\mu_{v_{i_{t}}}+\epsilon ⊙\sigma_{v_{i_{t}}}. \end{array}$$To perform link prediction, we compute the edge weights ${\hat{e}}_{ij_{t}}$ based on pair-wise distance defined by cosine similarity:(19)$$\begin{array}{cc} p({\hat{G}}_{t}|V_{t}) & =\prod_{i=1}^{n}\prod_{j=1}^{n}p\left({\hat{e}}_{ij_{t}}|v_{i_{t}},v_{j_{t}}\right)\\ {\hat{e}}_{ij_{t}} & =\sigma \left(\frac{\log w_{ij_{t}}-\log\left(1-w_{ij_{t}}\right)}{\tau }\right)\\ w_{ij_{t}} & =\frac{1}{2}\left(\frac{{v_{i_{t}}}^{T}v_{j_{t}}}{∥v_{i_{t}}∥\times ∥v_{j_{t}}∥}+1\right), \end{array}$$
where $\sigma \left(x\right)=1/\left(1+e^{-x}\right)$ is the sigmoid function and τ∈(0,1) is temperature. When τ→0, we have:(20)$$\begin{array}{c} \lim_{\tau \to 0}p\left({\hat{e}}_{ij_{t}}|v_{i_{t}},v_{j_{t}}\right)=0,\;if\;w_{ij_{t}}<0.5,\\ \lim_{\tau \to 0}p\left({\hat{e}}_{ij_{t}}|v_{i_{t}},v_{j_{t}}\right)=1,\;if\;w_{ij_{t}}>0.5, \end{array}$$
and $p({\hat{e}}_{ij_{t}}|v_{i_{t}},v_{j_{t}})$ becomes more sensitive when $w_{ij_{t}}$ is around 0.5. When τ=1, $p\left({\hat{e}}_{ij_{t}}|{v_{i}}_{t},{v_{j}}_{t}\right)=w_{ij_{t}}$. Our solution differs from previous works, generating edges such as: $p\left({\hat{e}}_{ij_{t}}|{v_{i}}_{t},{v_{j}}_{t}\right)=\sigma \left({{v_{i}}_{t}}^{T}{v_{j}}_{t}\right)$. Although both procedures use a continuous distribution to approximate the Bernoulli distribution, the logarithm and the normalization provide more stable gradients. With a proper temperature τ, the objective function is smoothed with a well-defined gradient $\frac{\partial {\hat{e}}_{ij_{t}}}{\partial w_{ij_{t}}}$. This is similar to the binary concrete distribution, but we do not involve the reparameterization trick and sample from the distribution, and the formula is slightly different. We provide this scheme as a better alternative to generate connectivities from a set of vectors.

Moreover, when τ→∞, the weight ${\hat{e}}_{ij_{t}}$ tends to be irrelevant to $w_{ij_{t}}$ as:(21)$$\lim_{\tau \to \infty }p\left({\hat{e}}_{ij_{t}}|v_{i_{t}},v_{j_{t}}\right)=0.5.$$This means that, when τ>1, this computation is similar to the VIB, which controls information from $w_{ij_{t}}$ to ${\hat{e}}_{ij_{t}}$ given a prior $q({\hat{e}}_{ij_{t}})=0.5$. Nevertheless, a focal solution for ${\hat{G}}_{t}$, such that ${\hat{e}}_{ij_{t}}$ stays around 0 or 1, may be more informative about the data, since each functional connectivities may distribute more separately. Hence, here, we study τ∈(0,1).

#### 3.3.2. Decoder

We design $q(Y^{\left(i\right)}|\hat{G})$ as a combination of a sequential graph-level GNN $p_{\omega_{0}}\left(r_{t}|{\hat{G}}_{t}\right)$ parameterized by $\omega_{0}$, and a multi-layer neural network $MLP_{\omega_{1}}$ parameterized by $\omega_{1}$, where $r_{t}\in R^{d}$ is the graph-embedding vector for ${\hat{G}}_{t}$. We implement $p_{\omega_{0}}\left(r_{t}|{\hat{G}}_{t}\right)$ based on a 1-layer Graph Isomorphism Network (GIN) [53]:(22)$$\begin{array}{cc} {X_{g}}^{\prime } & =MLP_{\eta }\left(\left({\hat{A}}_{t}+\left(1+\gamma \right)I\right)X_{g}\right)\\ r_{t} & =sum({X_{g}}^{\prime }), \end{array}$$
where $MLP_{\eta }$ is a multi-layer neural network parameterized by η, γ is a hyperparameter, $X_{g}\in R^{n\times d}$ is the matrix for *n* d-dimensional static variables defined by the delta function, ${\hat{A}}_{t}$ is the dynamic adjacency matrix, and sum is a summation across nodes serving as the READOUT function. Without a loss of generality, we treat static variables $X_{g}$ as learnable parameters, optimized jointly with the model.

Finally, we map the graph-level representation to the output behaviors via a multi-layer neural network $MLP_{\omega }$:(23)$$y_{t}=MLP_{\omega }\left(r_{t}\right).$$

We do not utilize time information in the decoder explicitly. Instead, we decode output behaviors $y_{t}$ from graphs ${\hat{G}}_{t}$ for every time step *t* separately, since the time-dependent dynamics are already extracted by $RNN_{\theta }$.

### 3.4. Differences from Existing Work

In this subsection, we briefly discuss the differences between our work and existing dynamic graph-learning methods based on optimization.

It is of great interest to learn the underlying graph structure of observed multivariate time series data. Chepuri et al. [54] propose a general learning scheme for noisy signals with a prior smoothness. Specifically, this learning approach finds denoised signals and a *K*-sparse graph structure from the data via optimizing:(24)$$\begin{array}{cc} \underset{\hat{X},\hat{A}}{\arg\min} & {∥\hat{X}-X∥}^{2}+\gamma tr({\hat{X}}^{T}L(\hat{A})\hat{X})\\ s.t. & \sum_{i,j}{\hat{e}}_{ij}=K, \end{array}$$
where $X\in R^{N\times T}$ is the noisy data, $\hat{X}$ is the denoised signals, $L(\hat{A})$ is the graph Laplacian matrix given the structure $\hat{A}$, and γ and *K* are hyperparameters. To generalize this learning procedure to dynamic graphs learning to study DFC, Jiang et al. [42] find the solution based on:(25)$$\begin{array}{cc} \underset{\left\{{\hat{x}}_{t}\right\},{\hat{A}}_{t}}{\arg\min} & \sum_{t=1}^{T}\left[{∥{\hat{x}}_{t}-x_{t}∥}^{2}+\gamma_{1}tr({\hat{x}}_{t}^{T}L({\hat{A}}_{t}){\hat{x}}_{t})\right]+\gamma_{2}\sum_{t=1}^{T-1}{∥{\hat{A}}_{t}-{\hat{A}}_{t+1}∥}_{1}\\ s.t. & \sum_{i,j}{\hat{e}}_{ij_{t}}=K, \end{array}$$
where $x_{t}\in R^{N}$ is the noisy signals on time step *t*, and the additional constraint serves as temporal smoothness. One should notice two important differences between the optimization approaches and VDGLVM, namely, the modeling goal and the learning framework.

First, we define the graph structure on Gaussian latent variables $V_{t}\in R^{n\times d}$ inferred from the observed data $x_{t}$, rather than on the data directly. The interest of the present work is not learning correlations of the observed signals, but finding probable dynamic communications between unseen latent variables underlying the observed data to interpret latent dynamics by LVMs. Since variables for DFC are less than the dimension of the data (n<N), we find the variables via a sequential VAE and perform link prediction on the variables to infer a graph from the data. This is why we introduce a dynamic graph as the latent variable and approximate the intractable posterior distribution with graph generative models. This is the most predominant difference.

Second, the learning framework is different. The optimization approach attempts to find the graph structure and eliminate additive Gaussian noise directly. To find the solution, the optimization framework requires varying forms of prior assumptions to constrain the solution space of the optimization problem. Consequently, it relies heavily on the formulation of the regularization terms, reflecting simplified prior knowledge of graph signals. Moreover, as indicated in Jiang et al. [42], computational complexity may be increased considerably with a more accurate modeling analysis. On the other hand, deep generative models can learn sophisticated mappings to find a more informative graph structure for its expressiveness and efficiency in a data-driven manner.

Nevertheless, the graph construction procedures are both based on the smoothness prior to the variables. $tr({\hat{X}}^{T}L(\hat{A})\hat{X})$ is the Dirichlet energy of a graph, given the structure $\hat{A}$, which encourages similar graph signals to be connected. It describes the distance between two signals as $e_{ij}{∥x_{i}-x_{j}∥}^{2}$, which is essentially consistent with our link-prediction scheme based on cosine similarity in Equation (Equation 19). Furthermore, the temporal smoothness priors are fundamentally consistent as well. While the optimization approach applies the assumption on the temporal graph property directly, VDGLVM merges the assumption in the RNN with an appropriate regularization on the parameters.

## 4. Experimental Setup

The validity and usefulness of our model depend on its performance when decoding behavior from neural spikes, and on whether the inferred representations preserve task-relevant structures. In this section, we first describe the datasets we use in the experiments. Then, we compare VDGLVM’s performance in decoding behaviors against other typical methods to demonstrate its feasibility. We further analyze qualitatively the representations that VDGLVM learned. Finally, we show the effects of two important hyperparameters on VDGLVM.

### 4.1. Dataset Description

We used two real-world neural datasets from the Neural Latents Benchmark [55] to demonstrate the interpretability of VDGLVM. All datasets are publicly available.

The primary dataset we conducted experiments on is Area2_Bump [56]. The data were recorded as a macaque was performing a center–out reaching task. Area2_Bump includes hand kinematics as behaviors and corresponding neural population spikes of neurons from area 2 of the somatosensory cortex.

We also used MC_Maze [57], and its scaled versions, MC_Maze_L, MC_Maze_M, and MC_Maze_S, as the secondary datasets to compare model performance on different datasets. The datasets consist of neural population spikes and the associated hand kinematics when a monkey made reaches with an instructed delay to visually presented targets while avoiding the boundaries of a virtual maze. The neurons were from the primary motor and dorsal premotor cortices, which directly involve motor control.

We used the recorded hand position as target behaviors and built models to map the neural spikes to the behaviors. We binned the ensemble spike activities into 20-ms bins.

### 4.2. Model Configurations

We used two unidirectional GRU layers as the feature extractor. The space of the features was 128-dimensional. We applied dropout with p=0.5 and layer normalization to avoid over-fitting. MLPs were implemented with two hidden layers. To further accelerate model training, we used ELU as the nonlinear activation function. In our model, we used three nodes for the graph, resulting in three dynamic edges. The static node features were in 128-dimensional space. We set the temperature τ=0.03 and VIB regularization multiplier $\beta =1\times 10^{-4}$ as defaults. The model was trained in a single Nvidia Titan RTX GPU for 256 epochs. We used the Adam optimizer with a learning rate of 0.001 to optimize the parameters. In our experiments, we split the dataset into 75% as the training set and 25% as the test set.

## 5. Results

### 5.1. Performance of Decoding Kinematics

We first wondered whether VDGLVM decodes behavior from neural spikes with a promised performance. We measured the performance by computing $R^{2}$ between the decoded behavior and normalized ground-truth behavior. To validate the expressiveness of VDGLVM, we assessed our model on different datasets and compared it against classic LVMs, including spike smoothing, GPFA, and SLDS. Since our model is based on nonlinear dynamical models and deep neural networks, it should significantly outperform these baselines. We also included a GRU-based RNN to compare, which is the backbone of VDGLVM without graphs and VIB. We trained these models in a supervised manner. The results are summarized in Table 1 (best results presented in bold font).

Spike smoothing, GPFA, and SLDS are essentially linear dynamical models. GPFA gives better results than spike smoothing because it accounts for the spikes across neurons and time in a probabilistic framework simultaneously. While both spike smoothing and GPFA benefit from the smoothness, SLDS uses a switching mechanism to approximate nonlinear dynamics with disparate linear dynamics and, thus, better models complex neural dynamics. RNN can capture more complex dynamics of nonlinear systems. Thus, we have the following observation from the table: in general, both RNN and VDGLVM outperform the methods based on linear models significantly. Another implicit advantage of RNN and VDGLVM is that they are end-to-end data-driven models, which are convenient to learn.

Furthermore, VDGLVM consistently achieves better performance than RNN. The predominant difference between VDGLVM and RNN is the design of the latent variable. This result proves the feasibility of utilizing graphs as the latent variable. It also implies that the introduced graph generative model and graph neural network may induce additional inductive bias. These modules ultimately improve expressiveness.

Deep neural networks tend to overfit. The results on MC_Maze and its scaled versions illustrate that both RNN and VDGLVM suffer from performance degeneration when samples are insufficient. However, VDGLVM is relatively alleviated, even though the model has extra trainable parameters for tackling graphs. Owing to the VIB framework, VDGLVM is less affected by noise and, thus, provides better generalization capability.

### 5.2. Latent Structure for Behavior Decoding

We are interested in the latent structure learned by VDGLVM. We include a three-dimensional latent space learned by RNN to compare, which represents existing LVMs based on a point process. Since we do not apply additional assumptions about the trajectories, the latent dynamics learned by VDGLVM should be similar to that by RNN. We visualize the dynamics after training VDGLVM and RNN on the Area2_Bump dataset with active trials in Figure 2.

Deep neural networks learn representations favorable to prediction. Since our main objective is decoding behavior from neural spikes, the actual dynamic functional connectivities may be further mapped to the representations favor for predictions. Therefore, as can be seen in Figure 2b,f, the latent trajectories learned by both VDGLVM and RNN preserve the task structure (reaching trajectories) in three-dimensional space.

However, the representations by RNN are only a low-dimensional embedding of the prediction, while the representations by VDGLVM contain a more informative structure. Looking at Figure 2a, we found that at least one dimension of the latent trajectories stays around 0 or 1 and provides no information about the dynamics. The results indicate that the dynamic relationships are local, governed by a subset of relationships. This sparsity agrees with the results of the constructed graph in previous work studying DFC [20]. The literature has shown that communication efficiency is one of the fundamental attributes that support neural function. In the language of complex networks, sparser connections between modules enable efficient inter-module integrations of information. We capture this by properly parameterizing the edge weights. The property is missed in most LVMs based on a point process, as shown in Figure 2e. Therefore, the dynamics graph inferred by our model provides a probable explanation, close to DFC, for latent dynamics by LVMs.

This illustrates that using a dynamic graph as the latent variable does associate latent dynamics with probable DFC. Meanwhile, as shown in Figure 2h, one may interpret the point process as a complete graph with node dynamics, where each node corresponds to one dimension of the trajectories. However, this trivial consideration does not have useful and practical insight into neural dynamics. On the contrary, as shown in Figure 2d, the graph by VDGLVM specifies an exact meaning for the dynamics as communications between nodes. One can conveniently associate the nodes with variables defined in the brain. Thus, we conclude that VDGLVM provides feasible neurophysiological interpretability to the inferred latent dynamics by existing LVMs.

### 5.3. Hyperparameter Tuning

We empirically analyzed two key hyperparameters by adjusting them and evaluating the resulting decoding performance.

#### 5.3.1. β in VIB Objective Function

Our model is based on the VIB framework. It restricts the information flow from the observations X to the latent variable $\hat{G}$ to minimize the VIB objective function. Given a proper β, this encourages the model to focus on information relevant to output Y in X. Meanwhile, the model learns to drop useless information, such as noise. Therefore, compared with the vanilla auto-encoder, the VIB-based model improves the generalization performance. However, an inappropriately large β will be harmful to the model. Since VIB forces the latent coding towards a given prior, the bottleneck becomes so small that the information flow is blocked. With extremely large β values, the model learns to transform the distribution of input p(X) to the prior $q(\hat{G})$, which is completely uninformative for decoding.

The property of β has been extensively studied in the literature. We verified this by studying the relationship between β and model performance. We present the result in Figure 3a. We manually adjusted β from typical settings to illustrate its effects on the model performance revealed by $R^{2}$. As shown in the figure, when we use β⩾0, increasing β improves the decoding performance against the baseline. The network attains the best when $10^{-5}⩾\beta ⩾10^{-4}$. Nevertheless, with a larger β, more and more discriminative information about input and output are blocked from input to latent coding. Thus, the model becomes seriously degenerated.

#### 5.3.2. τ in Graph Generation

Another hyperparameter significant for the model is the temperature τ. As discussed above, we only study τ∈(0,1). A small τ makes the sigmoid function steeper; it tends to be a step function and, thus, is more sensitive for the input around 0.5. More mathematically, with a smaller τ, the function has a higher Lipschitz constant. Since we directly involve this term in the propagation process, the Lipschitz constant of the model increases accordingly. While a smaller τ extracts more stable graph structures, an improperly small τ may make the model over-confident and, thus, less robust. We studied the association between τ and model performance. We present the result in Figure 3b. We found that the model may become unstable when τ<0.01, as the objective function becomes hard to optimize. Moreover, necessary dynamics are severely suppressed and thus insufficient to decode the target behavior. Therefore, the model tends to learn a trivial solution for the objective function.

## 6. Conclusions

Latent variable models (LVMs) for modeling neural data fail to determine the exact meaning of the inferred latent dynamics, while the literature has shown that dynamic functional connectivities (DFC) may be a possible mechanism of governing cognitive or perceptual tasks. In the present work, we propose Variational Dynamic Graph Latent Variable Model (VDGLVM), a representation learning model improving existing LVMs by interpreting the latent dynamics as DFC. To accomplish this, we design our model based on the variational information bottleneck and propose the use of a dynamic graph as the latent variable. Our model provides a probable explanation for latent dynamics captured by LVMs based on neurophysiology. In addition to the behavior dataset tested in the paper, we hope that the proposed model has the potential to apply to the analysis of neural activities in other perceptive and cognitive processes.

## Figures and Tables

**Figure 1 entropy-24-00152-f001:**
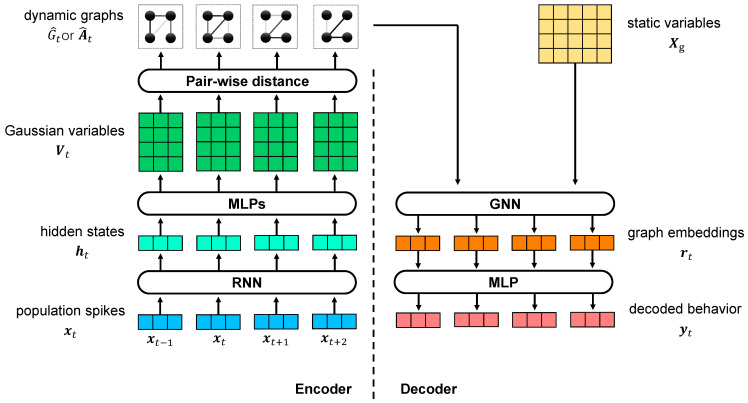
Illustration of VDGLVM using graphs to model the dynamic relationships between static variables. The left side is the encoder, consisting of a dynamical model extracting features from spikes data and a graph generative network building graphs for every time step. The right side is the decoder, including a GNN computing embeddings for graphs and a MLP projector mapping embeddings to behaviors.

**Figure 2 entropy-24-00152-f002:**
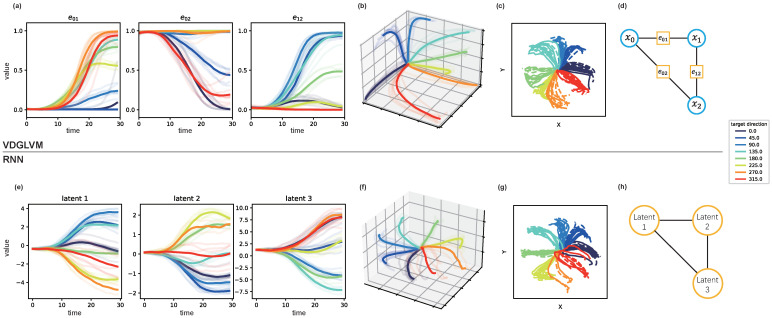
Inferred edge weights dynamics and behavior reconstructions by VDGLVM and inferred dynamics by RNN after training, where we use RNN to represent LVMs based on a point process. We visualize the edges evolving with time in (**a**). We combine them and show three-dimensional trajectories in (**b**), where each axis corresponds to an edge. We show behavior reconstructions for VDGLVM in (**c**), where the ground-truth trajectories are dash lines and decoding ones are solid lines. The graph with three dynamic edges and three nodes is shown in (**d**), where blue circles are static node variables and orange squares are dynamic edges. We show the three-dimensional trajectories inferred by RNN in (**e**,**f**). We show behavior reconstructions for RNN in (**g**). The equivalent graph structure of RNN is presented in (**h**), where orange circles are dynamic nodes.

**Figure 3 entropy-24-00152-f003:**
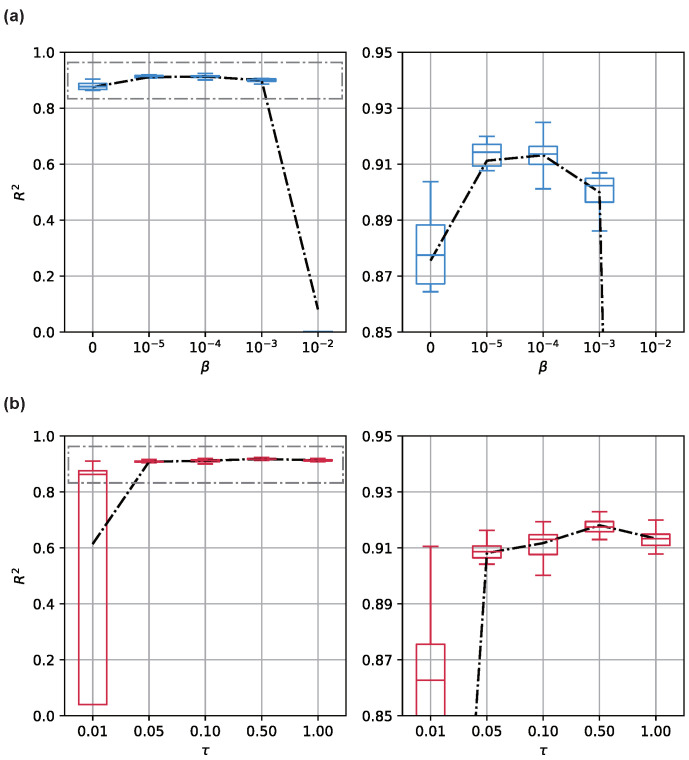
Decoding performance with the changing (**a**) β and (**b**) τ. Results are assessed with typical settings of the hyperparameters across 32 Monte Carlo simulations. The results are shown on the left side as two box plots, and the means are shown as lines. To present a more clear illustration, the results are zoomed in and shown on the right side.

**Table 1 entropy-24-00152-t001:** Comparison of different model performances with $R^{2}$.

Model	Area2_Bump	MC_Maze	MC_Maze_L	MC_Maze_M	MC_Maze_S
Smoothing	0.595	0.642	0.597	0.546	0.481
GPFA	0.613	0.669	0.598	0.571	0.533
SLDS	0.762	0.812	0.792	0.772	0.667
RNN	0.901	0.896	0.862	0.794	0.710
VDGLVM	**0.927**	**0.912**	**0.898**	**0.818**	**0.794**

## Data Availability

Publicly available datasets were analyzed in this study. The datasets can be found in https://gui.dandiarchive.org/#/dandiset (accessed on 5 December 2021).

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
