# Peer review of "Representation Learning for Dynamic Functional Connectivities via Variational Dynamic Graph Latent Variable Models"

_entropy, 2022, doi:10.3390/e24020152_

Round 1
Reviewer 1 Report
While the idea may appear natural as each Neural snapshot reading of the brain is indeed a complex graph that is evolving with time (see Bo Jiang et al.), the innovation is in treating the graphs themselves as latent entities.
The inpiration is good, but this reviewer felt there was a let-down since,
a. No elaboration on the behavioral data supposedly used for achieving an encoder/decoder system.
b. No real data was tested against to capture the validity of the paper.
I think, overall the paper brings a perspective that is publishable but some additional work is required (per (a) and (b)) and also comparing it to the dynamic models proposed in (Bo Jiang et al.) at least theoretically (they used real data for validation).
1. B. Jiang, Y. Huang, A. Panahi, Y. Yiyi, H. Krim and S. Smith.Dynamic Graph Learning: A Structure-Driven Approach, Jourmal of mathematics, Vol 9, No. 2, pg. 169, 2021 (MDPI).
Author Response
Thank you for your time and your inspiring comments. We believe that you have a misunderstanding due to our writing. We improved the paper by seriously considering your comments. We have rewritten some paragraphs to better support our point. To clear your confusion, please check the attachment for our response to your comments.

Reviewer 2 Report
This paper introduces dynamic graph as the latent variable and develops Variational Dynamic Graph Latent Variable Model (VDGLVM), which utilizes a graph generative model and a graph neural network to capture dynamic communication between nodes. The results are compared in terms of the inferred dynamics and the reconstruction. Some questions:
1. How to interpret the difference in Figures b and f? Which is better?
2. How does the proposed model generalize to other similar test datasets? Will the underlying structure be preserved?
3. What is the 'novel interpretability of the model? This should be discussed more in detail.
Author Response
Thank you for your time and your affirmation. Please check the attachment for our response to your questions.

Reviewer 3 Report
The manuscript discusses a new kind of latent variable model, variational dynamic graph latent variable model. This model is used to study the functional connectivity of spiking neuron populations, as well as the relationship of activity patterns with behavior. The authors propose to use a dynamic graph as the latent variable; this makes it possible to simulate dynamic processes in the communication of nodes.
The manuscript is original, interesting, and well structured. However, there are a number of comments both of a scientific and proofreading nature.
Comments
1. P.1, lines 13-14: “We show that our model provides guaranteed behavior decoding performance and novel interpretability for neural spikes.” And P. 2, lines 43-45: “We leverage the evolution of relationships between nodes in the graph to model population spiking and behavior, which may introduce novel interpretability for representation learning on neural data.”
Dear authors, could you describe and explain this announced result in more detail in one of the main sections of the article and in the conclusion.
2. P.3, line 90: “E = V × V” – “E \subseteq V × V”?
3. P.11, Fig.2: The color charts need legends. It should be written explicitly what the lines of different colors mean for each edge eij; the axes should also be labeled (for figures (a) and (f)).
Besides the points given above, it should be noted that the article needs proofreading. Some comments (the list is incomplete)
- P. 1, line 33: “(DFC) may a more accurate representation…” – “(DFC) may be a more accurate representation…”
- P. 2, lines 55, 56: “…where the elements li,t of L is the firing rate of the neuron i at time t. In practice, li,t is parameterized…” -- “…where the elements li,t of L are the firing rate of the neuron i at time t. In practice, li,t are parameterized…”
- P. 2, line 68: “inefficienct,” – misprint and extra comma: “inefficient”.
- P. 3: “huamn pose” – human pose.
- P. 3: “Generally, GNNs leverage a message passing [41] procedure to compute, including three steps.” – reformulate, please.
- P. 3, lines 94-95: “graph regression [46], and link prediction [47], etc.” – “graph regression [46], and link prediction [47], etc.”
- P. 6: “insantiate” – “instantiate”.
- P. 8, line 151: “Our solution is different from previous works…” – “Our solution differs from previous works…”.
- Ibid. “edeges” – “edges”.
- P. 8, line 162: “seperatately” – “separately”?
- P. 8, line 169: “We do not explicitly utilize time information in the decoder.” – “We do not use time information in the decoder explicitly”.
- P. 11, line 275: “unuseful” – “useless”.
- P. 12, line 299: “We could find” – “We found”.
- P. 12, line 305: “and behavior, In the present work…” – “and behavior, in the present work…”.
The list is not exhaustive.
Dear authors, I hope these comments will help you to improve the article.
Author Response
Thank you very much for your time and constructive comments. Please see the attachment to check our response and the details.

Round 2
Reviewer 1 Report
I made "one main comment" about the fact that this paper grounds their contribution on a dynamic graph model and asked them to specifically highlight the distinction of their paper with the paper cited, and they responded in the rebuttal and did nothing in the paper, and did not even mention the paper.
I am therefore at exactly the same point I was at previously, they HAVE NOT ANSWERED THE DISTINCTION to the community. I am therefore not giving this revision a positive (as I had suspected, given the quick response) support.
Author Response
Thank you for your time and your comments. Please check the attachment for our response and the latest version of our paper.

Round 3
Reviewer 1 Report
I think a sentence should be inserted stating the different goal of the paper should be stated and the optimization functional should be contrasted to to those proposed prior (e.g. [42]).
Author Response
Thank you for your time and your affirmation. Please check the attachment for our response.
